# ROYAL SOCIETY
# OPEN SCIENCE

# Research

chemical engineering

pre-coking, ZSM-5 zeolites, methanol to propylene, catalyst

**Author for correspondence:**
Dianhua Liu
e-mail: dhliu@ecust.edu.cn

# The guiding role of pre-coking on the coke deposition over ZSM-5 in methanol to propylene

Lin Wang[1,2], Jing Qi[2], Hongqiao Jiao[2], Liangcheng An[2], Chong Guan[2], Xiaojing Yong[2], Zhengwei Jin[2], Angui Zhang[2] and Dianhua Liu[1]

[1]State Key Laboratory of Chemical Engineering, East China University of Science and Technology, Shanghai 200237, People's Republic of China
[2]Coal to Liquids Chemical Research and Development Centre, Shenhua Ningxia Coal Industry Group, Yinchuan 750411, People's Republic of China

LW, 0000-0002-2494-4300

Deposition of carbonaceous compounds was used to improve the propylene selectivity of ZSM-5 by deactivating some acid sites meanwhile maintaining the high activity for methanol conversion. The carbonaceous species of pre-coked samples before and after MTP reactions were investigated by elementary analysis and thermogravimetric analysis (TGA). The results showed that pre-coke formed at low temperature (250°C) was unstable and easy to transform into polyaromatics species at the high reacting temperature, while combining 5% pre-coking process with 95% steam treatment at high temperature (480°C) was effective in inhibiting the formation of coke deposits and presented a significant improvement in the propylene selectivity.

## 1. Introduction

Propylene, as an important raw material, can be used to produce various products such as polypropylene, acrolein, acrylic acid and so on [1–3]. Propylene was mainly made from fossil resources via fluid catalytic cracking and steam cracking [4,5]. As one of the most important reactions in C1 chemistry, the methanol to propylene (MTP) process has the potential to play an increasing role in global chemicals manufacture. It has gained much attention these years, because of producing propylene with high selectivity via methanol from natural gas, coal, shale gas and biomass [6–10]. In order to understand the MTP process (prolonging the lifetime of catalyst, raising the catalyst activity, getting the best distribution of products), various studies of the MTP reaction have gained

considerable attention of many researchers in the past decades [11–16]. Now the silica–alumina zeolite ZSM-5 catalyst with MFI-type topology and high Si/Al ratio has been proved to be one of the most effective catalysts in typical MTP process, because of its unique three-dimensional pore structure (intersecting zigzag channels ($5.1 \times 5.5$ Å) and straight channels ($5.3 \times 5.6$ Å)), high propylene yield and anti-coking performance compared with SAPO-34 in a methanol to olefin process [17–20]. Whereas, the initial high activity of ZSM-5 results in high yield of gasoline and liquid gas [21], which decreases its commercial values.

It is, therefore, of great importance to find methods that can modify the acidity of ZSM-5 by chemical post-synthesis modification of zeolite frameworks as to decrease the side reactions of MTP [22–25]. To avoid side reactions, various modifications have been made. Among them, increasing framework Si/Al atomic ratio, i.e. by dealumination is effective, the general methods of which include extraction of framework aluminium by chemical agents, hydrothermal dealumination of zeolite frameworks and isomorphous substitution of framework silicon for aluminium [26–28]. Nevertheless, dealumination generally results in lattice deficiencies and the irreversible deactivation [29–31]. By contrast, there is no site poisoning by coke and catalyst deactivation due to coke is reversible. Coking can be expected to deactivate the acid sites and then affect the hydrogen transfer reactions depending on acid site density to suppress undesired reactions. Meanwhile, it also has obvious influence on the pore structure of molecular sieve: (i) limitation of the access of n-heptane to the active sites, (ii) blockage of the access to the sites of the cavities (or channel intersections) in which the coke molecules are situated, (iii) blockage of the access to the sites of the pores in which there are no coke molecules [32–35]. However, pre-coking using carbonaceous compounds with large molecules, it is supposed that coke only deposits externally, the internal sites keep unchanged [36]. These coke molecules trapped in the zeolite micropores, being relatively simple, are not generally inert with respect to the reactants or intermediates of the desired reactions and hence can significantly affect the activity and selectivity and be used as one of the surface modifications in some articles [37–40]. Bauer *et al.* suggested that, in terms of surface acidity inactivation, samples modified by carbonaceous deposits were found to be more effective compared to those by one-cycle silica deposition: (i) pre-coked HZSM-5 showed a higher shape selectivity during xylene isomerization, (ii) pre-coking of HFER had no specific effect on isobutene selectivity [38]. While other researchers found that catalyst with 0.6 wt% coke deposition presented 20 times the conversion on the fresh catalyst in chloromethane to olefins reaction, arising from the behaviour of coke working as an important reaction centre for olefin assembly, which can eliminate the induction period of the reaction and govern the conversion and product selectivity [40].

Regarding of the pre-coking, it is believed that a large amount of coke does not only selectively deposit on the outer surface of the catalyst, but also deactivate the internal active sites [41]. Thus, the pre-coking process should be carefully controlled and the growth rule of pre-coke species during the reaction should be mastered. Studies focusing on the stabilization or evolution of carbonaceous compounds on pre-coked catalysts during the reaction are few, thus the industrial application of pre-coking is worth deliberating.

In this study, various pre-coked catalysts were prepared with different reaction parameters: coking precursor, coking temperature and time. The textural properties of HZSM-5 catalyst before and after pre-coking have been characterized by series methods. The pre-coked catalysts were used in MTP reactions. The nature of coke species on the pre-coked ZSM-5 and the secondary-coked ZSM-5 in MTP conversions was examined by thermogravimetric analysis (TGA) method. Furthermore, the catalytic performances on propylene selectivity of these pre-coked catalysts have been investigated.

## 2. Experimental procedure

### 2.1. Materials and pre-coking procedures

The fabrication process of shaped parent ZSM-5 catalyst involves two key steps: the preparation of zeolite powder and shaping process of binder/zeolite. The obtained product was cylindrical, with size of 1.2–1.5 mm.

The ZSM-5 zeolite powder ($n_{\text{SiO}_2/\text{Al}_2\text{O}_3} = 180$) was bought from Suzhou Zhi hydrocarbon New Material Technology Co., Ltd. This powder was washed and filtrated repeatedly three times and then dried at 110°C overnight. Finally, the product was calcined at 550°C for 6 h in air.

Zeolite powder (350 g), binders (150 g, 75 wt% boehmite), sesbania powder (9 g), nitric acid (18 g, 66 wt%) and deionized water (180 g) were strongly blended at ambient temperature to form a homogeneous mixture, shaped into a uniform body by rapid extrusion moulding and then dried at 120°C for 12 h. The product was denoted as C-0.

Catalysts that used methanol (99.9 wt%) and 1-hexene (99.9 wt%) as the pre-coke precursors were named M-x-y and H-x-y, respectively, while catalysts that used methanol and deionized water as the pre-coking feed were named nM-W-x-y; x represents the coking time (h), y is the coking temperature (°C), and n is the mass per cent of methanol in the methanol and deionized water solution (%). Water is very important in MTP reaction. And during the reaction, MeOH produces water. The water produced by MeOH can also reduce partial pressure of methanol, remove heat of reaction and inhibit the formation of carbon deposits on catalysts.

The pre-coking ZSM-5 samples were prepared under different coking precursors (methanol, 1-hexene, and methanol and deionized water) with a weight hour space velocity (WHSV) of $1.0\,h^{-1}$, different temperature (250°C and 480°C) and time (8 h and 35 h) in a stainless-steel tube fixed-bed reactor (i.d. 12 mm). Before entering the reactor, the coking precursors need to be metered by the pump, then heated and gasified in turn. Before and after the pre-coking procedures, the catalysts were dried under the atmosphere of $N_2$ at 120°C for 3 h.

## 2.2. Characterization of catalysts

The BET surface area and porosity of the parent ZSM-5 and pre-coked catalysts were measured by $N_2$ adsorption and desorption method (Micromeritics ASAP 2400), the external surface area was tested by t-plot, and the mesopore volume was evaluated from the total pore volume minus the micropore volume. Prior to all measurement, the catalyst was outgassed at 120°C for 5 h in order to desorb water.

Temperature programmed desorption with ammonia gas ($NH_3$-TPD) was used to examine the acid sites amount and distribution. The catalyst was heated at 550°C for 1 h under helium atmosphere to remove physically absorbed water, and then cooled to 100°C and kept for 30 min for ammonia adsorption. The physically adsorbed ammonia was removed by blowing helium for 1 h, and then the catalyst was carried out to desorb chemically absorbed ammonia by increasing the temperature linearly from 100°C to 550°C with heating rate of $10°C\,min^{-1}$. In order to ensure the repeatability of data, all the operations are carried out under helium atmosphere.

Pre-coked catalysts were characterized by elementary analysis (Thermo Finnigan Flash EA 1112) and thermogravimetric analysis (TGA). The catalyst was placed in a crucible, lifted in the middle of the reactor tube. Before testing, the carrier gas $N_2$ was introduced at 120°C with a flow rate of $30\,ml\,min^{-1}$ for 4 h to remove the absorbed water in the catalyst. Then the carrier gas was replaced by a mixture of $O_2$ (80 vol%) diluted in nitrogen with a total flow rate of $50\,ml\,min^{-1}$, and the reactor temperature was increased from 120°C to 850°C with a heating rate of $10°C\,min^{-1}$.

## 2.3. Catalytic measurements

The catalytic testing was performed in a stainless-steel tube fixed-bed reactor (i.d. 12 mm). Five grams of the ZSM-5 zeolite catalyst were mixed with 5 g glass beads. The reaction was carried out at 480°C, and a WHSV of $1.0\,h^{-1}$. The methanol partial pressure was kept at 24 kPa by diluent $H_2O$. The product was analysed by gas chromatography with PLOT-Q column (30 m × 0.32 mm × 25 μm) from 120°C (keep 3 min) to 250°C (keep 15 min) with a heating rate of $20°C\,min^{-1}$.

# 3. Results and discussion

## 3.1. Physical characterization of the catalysts

The textural properties of pre-coked and parent catalysts are listed in table 1. It can be seen that most of the catalysts' surface area and micropore volume are decreased after pre-coking, except for sample H-8-250, whose pre-coking species came from the conversion of 1-hexene at 250°C. For pre-coked samples (1M-W and 5M-W) prepared with abundant water at high temperature (480°C), an increasing trend of mesopore volume can be seen as the water concentration increased, suggesting the dealumination effect of steam on the catalysts, while the effect of coke on blocking the micropores can be observed, as the reduction trend of micropore volume did not follow the steam content. The pore volume of pre-coked sample 5M-W-35-480 is lower than sample 1M-W-35-480 and is similar to present sample C-0.

For pre-coked catalysts using methanol as coke precursor, the external surface is decreased notably as the cooking time extended from 8 h to 35 h at 480°C, while no distinguishing trend can be seen in the micropore volume, indicating that the pre-coke species mainly deposited on the outer surface of the

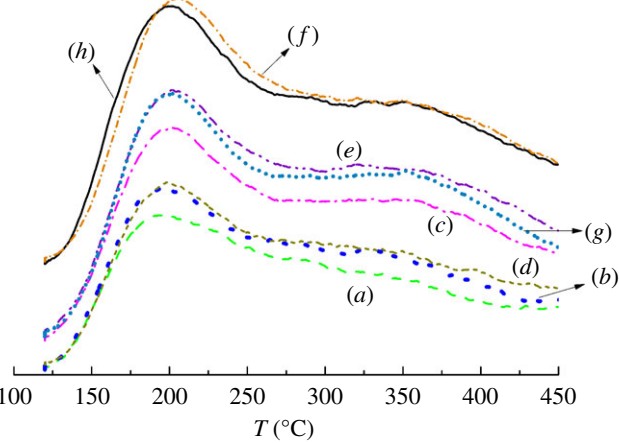

**Figure 1.** $NH_3$-TPD profiles of parent and pre-coked samples: (*a*) 1M-W-35-480, (*b*) 5M-W-35-480, (*c*) M-8-480, (*d*) M-35-480, (*e*) M-35-250, (*f*) H-8-250, (*g*) H-8-480 and (*h*) C-0 (present ZSM-5) [42].

**Table 1.** Textural properties of parent and pre-coked ZSM-5 [42]. $S_{surf}$: specific surface area.

| sample | $S_{surf}$ /(m² g⁻¹) | | | pore volume/(cm³ g⁻¹) | | |
|---|---|---|---|---|---|---|
| | BET[a] | micro[b] | external[b] | micro[b] | meco[c] | total[d] |
| C-0 | 349 | 302 | 47 | 0.146 | 0.231 | 0.377 |
| 1M-W-35-480 | 321 | 276 | 45 | 0.140 | 0.250 | 0.390 |
| 5M-W-35-480 | 319 | 271 | 48 | 0.135 | 0.235 | 0.370 |
| M-8-480 | 329 | 279 | 50 | 0.137 | 0.237 | 0.374 |
| M-35-480 | 318 | 276 | 42 | 0.137 | 0.225 | 0.362 |
| M-35-250 | 340 | 293 | 47 | 0.142 | 0.232 | 0.374 |
| H-8-250 | 538 | 445 | 93 | 0.219 | 0.408 | 0.627 |
| H-8-480 | 333 | 291 | 42 | 0.140 | 0.229 | 0.369 |

BET[a]: multi-point BET surface area.
[b]from t-plot method.
[c]from BJH method.
[d]at P/P0 ≈ 0.98.

particles. For pre-coked sample (M-35-250) prepared at low temperature (250°C), a lesser destruction of micropores is observed, and the external surface and mesopore volume are nearly unchanged, which can be attributed to the low temperature limiting the forming of carbonaceous compounds.

Catalyst using 1-hexene as pre-coke species has lower external surface compared with catalyst using methanol as coke precursor at the same reaction conditions. This is due to the fact that a large molecule is more likely to be condensed and deposited on the external surface of the zeolite [43].

## 3.2. Acid properties of the pre-coked catalysts

The $NH_3$-TPD profiles obtained from parent and pre-coked ZSM-5 samples are shown in figure 1. Two broad desorption peaks with maxima at 202°C and 315°C are assignable to weak acid site caused by ammonium cations and strong acid sites, respectively [44].The desorption peaks in pre-coked sample H-8-250 are no different from present ZSM-5, indicating that carbonaceous compounds formed by 1-hexene at low temperature (250°C) are really unstable, and can be easily decomposed at high temperature; thus, this kind of pre-coke species had no effect on the acidity of catalysts.

The $NH_3$-TPD results reveal only the difference in the amount of acid sites but not in the strength of the pre-coked catalysts. The acidity distributions of samples M-35-250 and H-8-480 are almost the same, though they were prepared under quite different pre-coking conditions. A similar finding was obtained

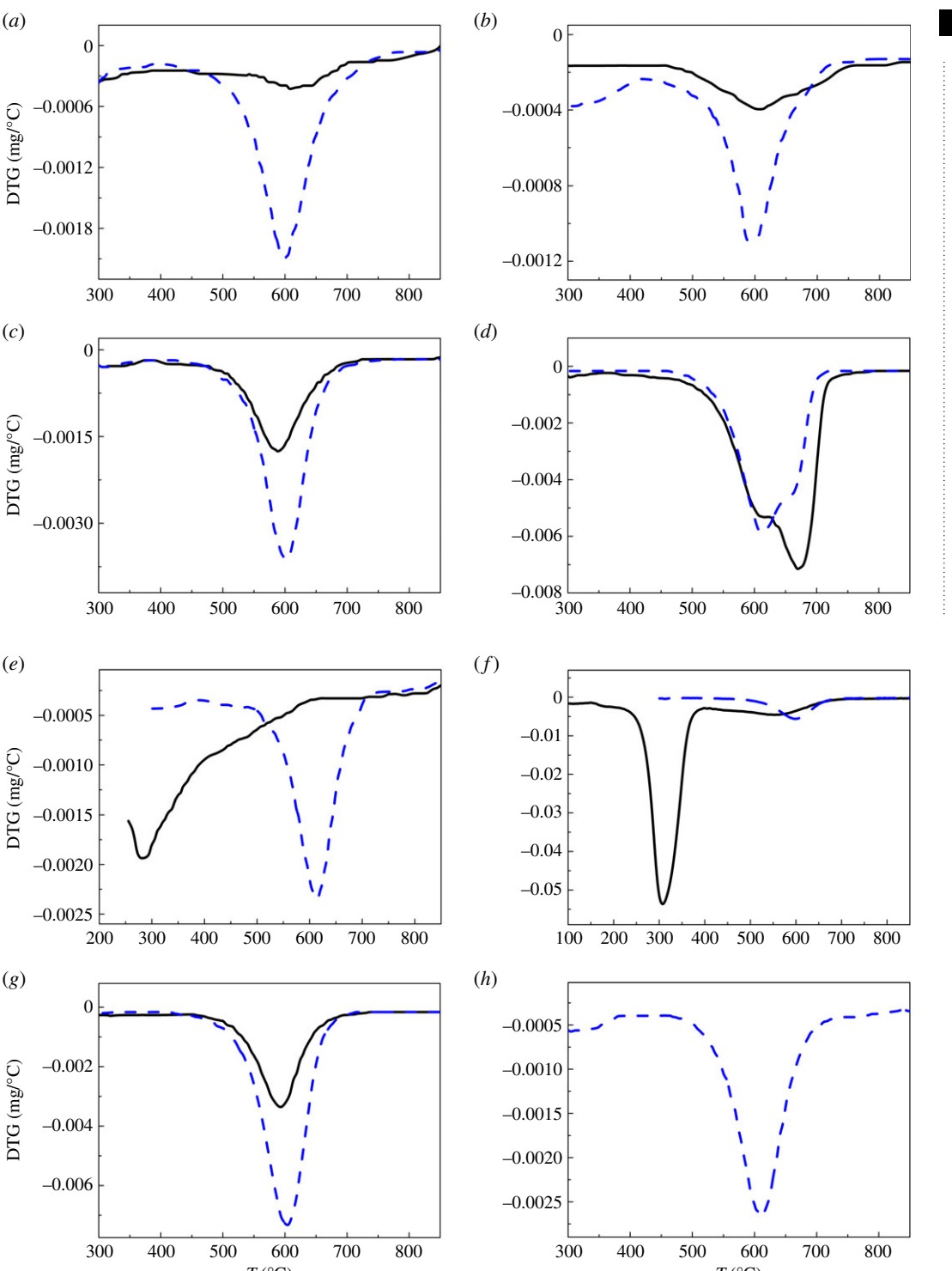

**Figure 2.** Differential thermogravimetry (DTG) profiles of pre-coked catalysts before (solid line) and after (dashed line) MTP reaction: (a) 1M-W-35-480, (b) 5M-W-35-480, (c) M-8-480, (d) M-35-480, (e) M-35-250, (f) H-8-250, (g) H-8-480 and (h) C-0 (present ZSM-5) [42].

for the samples 5M-W-35-480 and M-35-480. This observation indicates that different coordinated pre-coking parameters are possible to result in the same modification of the acidity.

## 3.3. Coke characterization of the pre-coked catalysts before and after MTP reaction

The amount of oxidable coke was determined by TGA, and the results for these pre-coked catalysts are shown in figures 2 and 3. It can be seen in figure 2 that burning off the pre-coke formed at high

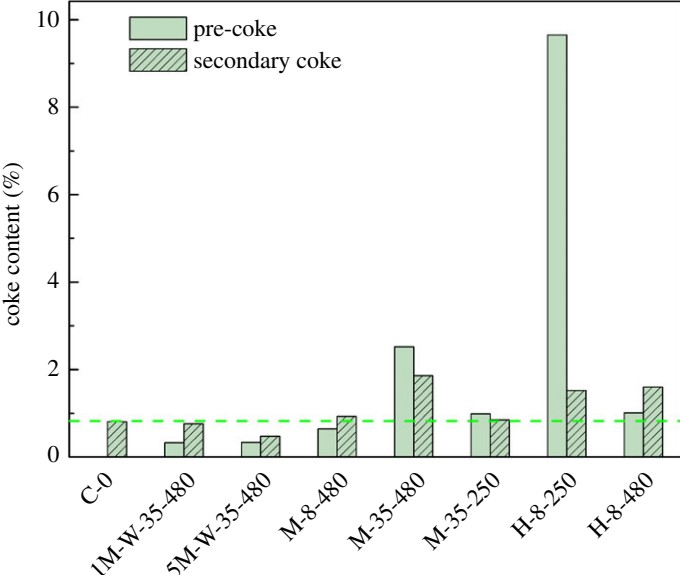

**Figure 3.** Comparison of pre-coke content and secondary-coke content of each sample [42].

**Table 2.** Elementary analysis of carbonaceous compounds of pre-coked catalysts before and after MTP reaction [42].

| samples | C/H mass ratio of pre-coke | C/H mass ratio of secondary-coke |
|---|---|---|
| M-35-480 | 21.30 | 27.79 |
| H-8-250 | 8.61 | 14.50 |
| C-0 | 0 | 5.19 |

temperature (480°C) reveals a combustion peak at 600°C, which is attributed to the combustion of polyaromatic species. While the pre-coked sample M-35-480 is the exception, as two combustion peaks can be observed in figure 2*d*, revealing two types of coke species [45]: coke I (600°C) and coke II (670°C), and the amount of these coke species are exchanged after MTP reaction. This observation can be ascribing to the different coke locations of the pre-coke and secondary-coke [46]. Pre-coke on sample M-35-480 is deposited mainly on the external surface, while on the effect of steam during the MTP reaction, part of the pre-coke decomposed and reassembled at the internal pores [17], thus increasing the proportion of internal coke.

The combustion peak of pre-coke formed at low temperature (250°C) is centred on 300°C (figure 2*e,f*), containing mainly the olefinic compounds [47]. This kind of coke species is condensed and rearranged via hydrogen transfer steps at high temperature (480°C), resulting in the formation of polyaromatics. This fact reflects that coke formed at low temperature is of olefinic nature, but the stable form of coke at high temperature is polyaromatic, and is highly condensed and dehydrogenized olefinic or aliphatic species.

The elementary analysis of carbonaceous compounds of these abnormal pre-coked samples is shown in table 2. It can be seen that pre-coke formed at low temperature (250°C) via 1-hexene is of more olefinic nature, while the coke species turned to be of more aromatic nature after MTP reaction; this fact verified that coke formed at low temperature is unstable at high temperature [48], and will be transformed to polyaromatics. While pre-coke formed at severely high temperature via methanol conversion is more aromatic and condensed, this type of pre-coke is quite stable at high temperature, but continues to dehydrogenize and condense into carbonaceous compounds with larger size.

Figure 3 shows the coke content of pre-coked catalysts before and after MTP reaction. It can be seen that the coke content of pre-coked sample 5M-W-35-480 is obviously lower than present catalysts after MTP conversions. Secondary-coke content among samples 1M-W-35-480, M-8-480, M-35-250 and present sample C-0 show no distinguishable difference, while the others (M-35-480, H-8-250, H-8-480) are notably higher than sample C-0. This fact suggests that different pre-coking conditions lead to the different distribution of coke deposits and, furthermore, results in the different textural properties of the catalysts, taking charge of the various coke performances.

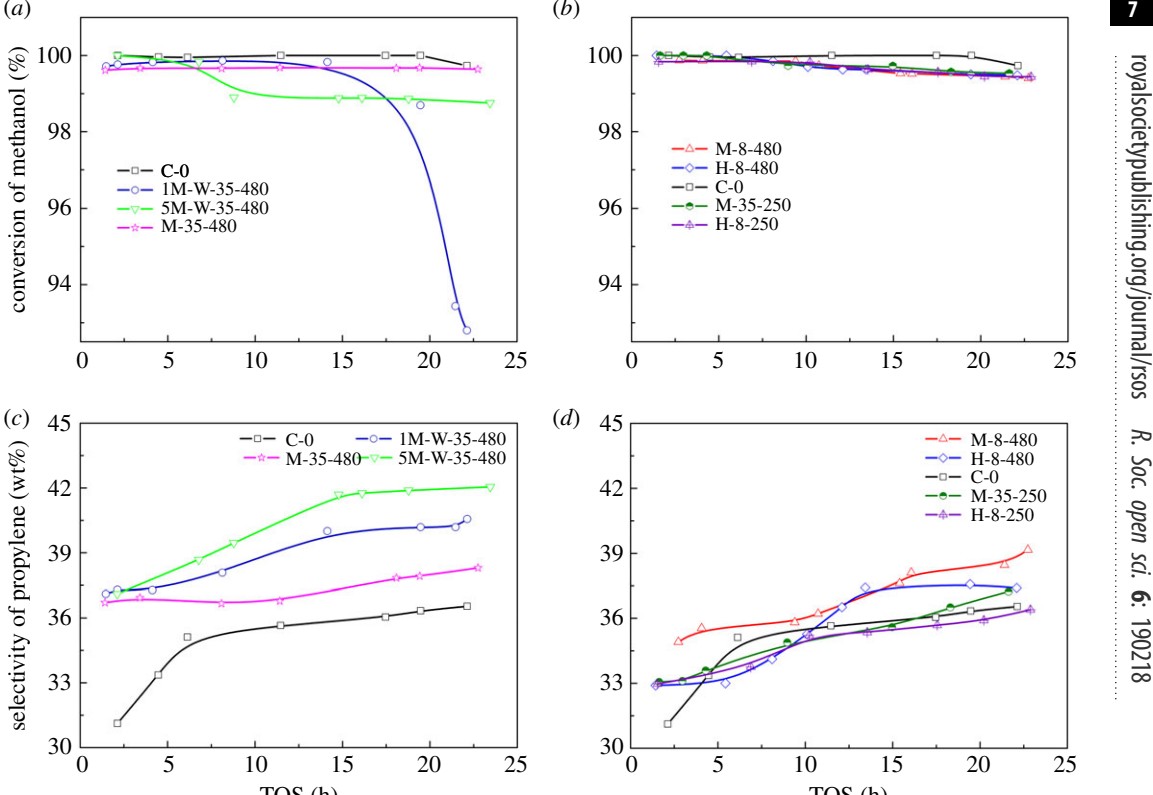

**Figure 4.** Catalytic performances of pre-coked ZSM-5 in MTP reaction: (*a,b*) methanol conversion; (*c,d*) selectivity of propylene [42].

Pre-coking via 1-hexene conversion obtains higher coke deposition compared with using methanol as pre-coke precursor. This can be attributed to the dehydration of methanol, which generating the equal moles of water and then suppressed the formation of coke [49]. No water appeared at the conversion of 1-hexene, thus the pre-coke in the pores or on the outer surface acts as coke-nucleus and accelerates the formation of secondary-coke.

Combining the pre-coking treatment with steaming is very effective to inhibit the secondary-coke formation, as both coke and water can be used to modify the pore structure and acidity of the catalyst [50], while the proportions of these two factors need to be optimized. It can be seen that pre-coking by 5% methanol and steam is more effective than 1% methanol and steam in suppressing the coke deposition. This can be clarified by the pre-coke distribution which was verified by the textural properties of these two pre-coked samples characterized by BET methods. Pre-coke is likely to fill in the mesopores formed by dealumination by steam, primarily, thus the acid sites near the mesopore walls are covered, and the diffusion resistance of molecules is strengthened by the steric hindrance of pre-coke, increasing the difficulty of macromolecular carbon deposition on the mesopores. Thus, suitable pre-coking process is instead done well in restraining coke deposition rather than hydrothermal treatment.

## 3.4. Catalytic properties of the pre-coked ZSM-5 in MTP reaction

The pre-coked catalysts were classified into two types according to the acidity of the catalyst. Samples 1M-W-35-480, 5M-W-35-480 and M-35-480 were prepared at relatively severe coking conditions, and the destruction of acid sites was more obvious; hence, these pre-coked catalysts were marked as type A, and the others were marked as type B. The conversion of methanol and selectivity of propylene are displayed against time on steam (TOS) and the results are shown in figure 4. The conversions of pre-coked catalysts are above 99%, except sample 1M-W-35-480, whose conversion decreased rapidly after 15 h of TOS. The reason is that there is too much water added during the preprocessing of the sample 1M-W-35-480. It causes serious dealumination when sample is under hydrothermal condition and causes many losses of acidic sites.

The propylene selectivity of pre-coked catalysts (type A, figure 4c) are substantially higher than pre-coked catalyst of type B (figure 4d), the propylene selectivity of sample 5M-W-35-480 reached 42% after

15 h of TOS. This is due to the coordinated modification of coke and steam, resulting in a suitable pore structure for the production of propylene. As sample M-35-480 was severely coked, the diffusion limitation of reacting molecules was enhanced, which not only restrained the production of macromolecules, but also affected the diffusion of micromolecules like propylene [51]. Pre-coking prepared at low temperature (250°C) presents no improvement in propylene selectivity, arising from the unstable form of the carbonaceous compounds.

# 4. Conclusion

The pre-coking modification of ZSM-5 zeolite has been performed to improve the propylene selectivity of the catalysts maintaining high activity. Detailed features of pre-coked catalysts have been obtained by the results of the characterization and catalytic investigation. Pre-coking had less destruction to the acid sites of ZSM-5 compared with steam, and retained the high activity of the catalyst. Pre-coke formed at low temperature (250°C) was unstable; the olefinic and aliphatic compounds were transformed into polyaromatic species at high reacting temperature. While 5% pre-coking combined with 95% steaming treatment at high temperature (480°C) was very effective to inhibit the coke formation during the MTP conversions, as the pre-coke primarily deposited on the mesopores formed by dealumination of the crystal, and the selectivity of propylene reached 42%.

Data accessibility. Data available from the Dryad Digital Repository: https://doi.org/10.5061/dryad.6c8sk60 [42].

Authors' contributions. L.W. designed this work, conducted the experimental operations and catalytic optimization, and drafted the manuscript; J.Q., H.J., L.A. and C.G. participated in the analysis and interpretation of data; X.Y., Z.J. and A.Z. revised the manuscript for important intellectual contents; D.L. ensured that the work was appropriately investigated and analysed. All authors gave final approval for publication.

Competing interests. We declare we have no competing interests.

Funding. This work was supported by the National International Cooperation S & T Project of China (grant no. 2015DFA40660), Key R&D Programs of NingXia Province (grant nos. 2018BDE63020, 2018BDE63003 and 2018BDE63007).

Acknowledgements. We thank Prof. Dr Siegfried Stapf for the analysis of data and revising the manuscript.

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
