## [Reviewer comments · Royal Society Open Science]

Review History

RSOS-190218.R0 (Original submission)

Review form: Reviewer 1

Is the manuscript scientifically sound in its present form?

Yes

Are the interpretations and conclusions justified by the results?

Yes

Is the language acceptable?

Yes

Is it clear how to access all supporting data?

Yes

Do you have any ethical concerns with this paper?

No

Have you any concerns about statistical analyses in this paper?

No

Recommendation?

Accept with minor revision (please list in comments)

Comments to the Author(s)

This work does some researches on the pre-coked ZSM-5 catalysts, which were used in the MTP reactions. The influence of the catalysts pre-coking has been studied and discussed.

The research methods of this study are common. However, the techniques in this study are very important in practice. And, the research area of the subject has broad range of interest. I suggest to accept this paper after the following issue has been addressed.

There are some revisions:

Introduction:

The description in the introduction parts about the influences in different pre-coking conditions and in different systems are confused. It needs rearrangement. Also, the language needs to be polished and proofread.

Experimental:

- During the experiment, water is one condition which is under controlled. But MeOH produces water during the reactions. What will this part of water affect the reactions?

- During the NH₃-TPD measurements the pre-coking catalyst has been heated to 550°C. Here the author does not mention in what gas condition. Will the choice of gas influence the measurements?

- In experimental part, some critical experimental details and standards are not mentioned. For example, what is the source and the purity of the MeOH and ZSM-5? What water has been used? Distilled or deionized?

Results and Discussion:

- In the section 3.1, it says "The pore volume of pre-coked sample 5M-W-35-480 is much lower than sample 1M-W-35-480". But in table.1, the pore volume difference between 5M-W-35-480 and 1M-W-35-480 are only 0.02.

- In the section 3.4, it says "The conversions of pre-coked catalysts are above 99%, except sample 1M-W-35-480, whose conversion decreased rapidly after 15 h of TOS." Why the conversion decrease? The author does not give explanation.

Language:

There are several errors in the manuscript. For example, "stream" and "streaming". These kinds of description are quite erratic. A manuscript should be proofread very carefully before the submission.

In summary, considering the subject of this manuscript has broad range of interest. After the proper revisions, the publication of this manuscript is recommended.

Review form: Reviewer 2

Is the manuscript scientifically sound in its present form?

Yes

Are the interpretations and conclusions justified by the results?

Yes

Is the language acceptable?

Yes

Is it clear how to access all supporting data?

Yes

Do you have any ethical concerns with this paper?

No

Have you any concerns about statistical analyses in this paper?

No

Recommendation?

Accept as is

Comments to the Author(s)

The authors carefully examined the effects of reaction parameters on the propylene selectivity of the pre-coked ZSM-5 based catalysts and found an efficient way to inhibit the coke formation during MTP conversions. The study is important and meaningful. The results are convincing. I recommend the publication of this manuscript as is.

Decision letter (RSOS-190218.R0)

25-Jun-2019

Dear Dr Wang:

Title: The Guiding Role of Pre-coking on the Coke Deposition over ZSM-5 in Methanol to Propylene
Manuscript ID: RSOS-190218

Thank you for submitting the above manuscript to Royal Society Open Science. On behalf of the Editors and the Royal Society of Chemistry, I am pleased to inform you that your manuscript will be accepted for publication in Royal Society Open Science subject to minor revision in accordance with the referee suggestions. Please find the reviewers' comments at the end of this email.

The reviewers and handling editors have recommended publication, but also suggest some minor revisions to your manuscript. Therefore, I invite you to respond to the comments and revise your manuscript. I apologise this has taken longer than usual.

Because the schedule for publication is very tight, it is a condition of publication that you submit

the revised version of your manuscript before 04-Jul-2019. Please note that the revision deadline will expire at 00.00am on this date. If you do not think you will be able to meet this date please let me know immediately.

Best wishes,
Dr Laura Smith
Publishing Editor, Journals

Royal Society of Chemistry
Thomas Graham House
Science Park, Milton Road
Cambridge, CB4 0WF

Royal Society Open Science - Chemistry Editorial Office

On behalf of the Subject Editor Professor Anthony Stace and the Associate Editor Professor Tobias Hertel.

RSC Associate Editor:
Comments to the Author:
(There are no comments.)

RSC Subject Editor:
Comments to the Author:
(There are no comments.)

Reviewer comments to Author:
Reviewer: 1

Comments to the Author(s)

This work does some researches on the pre-coked ZSM-5 catalysts, which were used in the MTP reactions. The influence of the catalysts pre-coking has been studied and discussed. The research methods of this study are common. However, the techniques in this study are very important in practice. And, the research area of the subject has broad range of interest. I suggest to accept this paper after the following issue has been addressed.

There are some revisions:

Introduction:

The description in the introduction parts about the influences in different pre-coking conditions and in different systems are confused. It needs rearrangement. Also, the language needs to be polished and proofread.

Experimental:

- During the experiment, water is one condition which is under controlled. But MeOH produces water during the reactions. What will this part of water affect the reactions?

- During the NH₃-TPD measurements the pre-coking catalyst has been heated to 550°C. Here the author does not mention in what gas condition. Will the choice of gas influence the measurements?

- In experimental part, some critical experimental details and standards are not mentioned. For example, what is the source and the purity of the MeOH and ZSM-5? What water has been used? Distilled or deionized?

Results and Discussion:

- In the section 3.1, it says "The pore volume of pre-coked sample 5M-W-35-480 is much lower than sample 1M-W-35-480". But in table.1, the pore volume difference between 5M-W-35-480 and 1M-W-35-480 are only 0.02.

- In the section 3.4, it says "The conversions of pre-coked catalysts are above 99%, except sample

1M-W-35-480, whose conversion decreased rapidly after 15 h of TOS." Why the conversion decrease? The author does not give explanation.

Language:

There are several errors in the manuscript. For example, "stream" and "streaming". These kinds of description are quite erratic. A manuscript should be proofread very carefully before the submission.

In summary, considering the subject of this manuscript has broad range of interest. After the proper revisions, the publication of this manuscript is recommended.

Reviewer: 2

Comments to the Author(s)

The authors carefully examined the effects of reaction parameters on the propylene selectivity of the pre-coked ZSM-5 based catalysts and found an efficient way to inhibit the coke formation during MTP conversions. The study is important and meaningful. The results are convincing. I recommend the publication of this manuscript as is.

Author's Response to Decision Letter for (RSOS-190218.R0)

See Appendices A & B.

Decision letter (RSOS-190218.R1)

12-Aug-2019

Dear Dr Wang:

Title: The Guiding Role of Pre-coking on the Coke Deposition over ZSM-5 in Methanol to Propylene
Manuscript ID: RSOS-190218.R1

It is a pleasure to accept your manuscript in its current form for publication in Royal Society Open Science. The chemistry content of Royal Society Open Science is published in collaboration with the Royal Society of Chemistry.

On behalf of the Subject Editor Professor Anthony Stace and the Associate Editor Professor Tobias Hertel.

RSC Associate Editor
Comments to the Author:
(There are no comments.)

Reviewer(s)' Comments to Author:

Appendix A

Dear editor:

On behalf of all authors, thank you very much for your time and patient in our manuscript entitled “*The Guiding Role of Pre-coking on the Coke Deposition over ZSM-5 in Methanol to Propylene*” (ID: RSOS-190218). We are extremely appreciated for your consideration of publication of our work in Royal Society Open Science.

All responsible comments are valuable and helpful for revising and improving our paper. We respond carefully to each of the issues raised by the referees, outlining all corrections and improvements made as follows. All changes or the corrections have been marked in red ink in the revised manuscript. In addition, another version of the manuscript, without any marked text, is also provided.

In the text below, comments by the referee are underlined; our response is in normal text. Text that has been added or edited was marked in red ink. We hope that these amendments are sufficient and that the revised manuscript can be accepted for publication.

Thank you for your expert and expeditious handling of this paper as always!

Very best regards,

Dr. Lin Wang

State Key Laboratory of Chemical Engineering, East China University of Science and Technology, Shanghai 200237, People’s Republic of China;

China Energy Group Ningxia Coal Industry Co., Ltd., Yinchuan 750001, People’s Republic of China.

Tel: +86 951 6963742, Fax: +86 951 6963779

E-mail: wanglin022506@163.com

Responses to the Reviewers

We appreciate all the valuable suggestions, revised portion are marked in red in the revised manuscript and our responses to the comments are as follows:

Referee: 1

Comments:

This work does some researches on the pre-coked ZSM-5 catalysts, which were used in the MTP reactions. The influence of the catalysts pre-coking has been studied and discussed.

The research methods of this study are common. However, the techniques in this study are very important in practice. And, the research area of the subject has broad range of interest. I suggest to accept this paper after the following issue has been addressed.

Question 1: Introduction:

The description in the introduction parts about the influences in different pre-coking conditions and in different systems are confused. It needs rearrangement. Also, the language needs to be polished and proofread.

Response: Thanks for the rigorous question from reviewer. The Introduction section has been rearranged and polished in the revised manuscript.

Question 2: During the experiment, water is one condition which is under controlled. But MeOH produces water during the reactions. What will this part of water affect the reactions?

Response: We agree with the suggestions from reviewer. Water is very important in MTP reaction. The water produced by MeOH can also reduce partial pressure of methanol, remove heat of reaction and inhibit the formation of carbon deposits on Catalysts. The Relevant content has been explained in the revised manuscript.

Question 3: During the NH₃-TPD measurements the pre-coking catalyst has been heated to 550 °C . Here the author does not mention in what gas condition. Will the choice of gas influence the measurements?.

Response: Thanks for the reminding from referee. In order to ensure the repeatability of data, all the operations are carried out under helium atmosphere. The Relevant content has been revised in the revised manuscript.

Question 4: In experimental part, some critical experimental details and standards are not mentioned. For example, what is the source and the purity of the MeOH and ZSM-5? What water has been used? Distilled or deionized?

Response: We agree with the suggestions from reviewer. It has been corrected in the revised manuscript.

Question 5: In the section 3.1, it says “The pore volume of pre-coked sample 5M-W-35-480 is much lower than sample 1M-W-35-480”. But in table.1, the pore volume difference between 5M-W-35-480 and 1M-W-35-480 are only 0.02.

Response: Thanks for the reminding from referee. It’s a misdescription and has been corrected in the revised manuscript.

Question 6: In the section 3.4, it says “The conversions of pre-coked catalysts are above 99%, except sample 1M-W-35-480, whose conversion decreased rapidly after 15 h of TOS.” Why the conversion decrease? The author does not give explanation.

Response: Thanks for the reminding from referee. The reason is there are too much water added during the preproccession of the sample 1M-W-35-480. It causes serious dealumination when sample is under hydrothermal condition and cause many losses of acidic sites. The Relevant content has been explained in the revised manuscript.

Question 7: There are several errors in the manuscript. For example, “stream” and “streaming”. These kinds of description are quite erratic. A manuscript should be proofread very carefully before the submission.

Response: We are deeply sorry for this careless mistake. the revised manuscript has been carefully polished.

Appendix B

The Guiding Role of Pre-coking on the Coke Deposition over ZSM-5 in Methanol to Propylene

Lin Wang^{1,2}, Jing Qi², Hongqiao Jiao², Liangcheng An², Chong Guan², Xiaojing Yong², Zhengwei Jin², Angui Zhang² and Dianhua Liu^{1,*}

¹State Key Laboratory of Chemical Engineering, East China University of Science and Technology, Shanghai 200237, China

²Coal to Liquids Chemical Research and Development Center, Shenhua Ningxia Coal Industry Group, Yinchuan 750411, China

*Corresponding author: Tel.: +86-21-64252151; E-mail: dhliu@ecust.edu.cn

Abstract: Deposition of carbonaceous compounds was used to improve the propylene selectivity of ZSM-5 by deactivating some acid sites meanwhile maintaining the high activity for methanol conversion. The carbonaceous species of pre-coked samples before and after MTP reactions were investigated by elementary analysis and Thermo Gravimetric Analyzer (TGA). The results showed that pre-coke formed at low temperature (250 °C) was unstable and easily to transform into polyaromatics species at the high reacting temperature. While combing 5% pre-coking process with 95% steam treatment at high temperature (480 °C) was effective in inhibiting the formation of coke deposits and presented a significant improvement in the propylene selectivity.

Key words: Pre-coking, ZSM-5 zeolites, Methanol to Propylene, Catalyst

1. Introduction

Propylene, as an important raw material, can be used to produce various products such as polypropylene, acrolein, acrylic acid and so on [1-3]. Propylene was mainly made from fossil resources via fluid catalytic cracking (FCC) and steam cracking [4, 5]. As one of the most important reactions in C1 chemistry, the methanol to propylene (MTP) process has the potential to play an increasing role in global chemicals manufacture. It has gained much attention these years, because of producing propylene with high selectivity via methanol from natural gas, coal, shale gas and biomass [6-10]. In order to understand the MTP process (prolonging the lifetime of catalyst, raising the catalyst activity, getting the best distribution of products), various studies of the MTP reaction have gained considerable attentions of many researchers in the past decades [11-16]. Now the silica-alumina zeolite ZSM-5 catalyst with MFI-type topology and high Si/Al ratio has been proved to be one of the most effective catalysts in typical MTP process, because of its unique three-dimensional pore structure (intersecting zigzag channels ($5.1 \times 5.5 \text{ \AA}$) and straight channels ($5.3 \times 5.6 \text{ \AA}$)), high propylene yield and anti-coking performance compared with SAPO-34 in a methanol to olefin process (MTO) [17-20]. Whereas, the initial high activity of ZSM-5 results in high yield of gasoline and liquid gas [21], which decreases its commercial values.

It is, therefore, of great importance to find methods that can modify the acidity of ZSM-5 by chemical post-synthesis modification of zeolite frameworks as to decrease the side-reactions of MTP [22-25]. To avoid side reactions, various modifications have been made. Among them, increasing framework Si/Al atomic ratio i.e., by dealumination is effective, the general methods of which include extraction of framework aluminium by chemical agents, hydrothermal dealumination of zeolite frameworks and isomorphous substitution of framework silicon for aluminium [26-28]. Nevertheless, dealumination generally results in lattice deficiencies and the irreversible deactivation [29-31]. In contrast, there is no site poisoning by coke and catalyst deactivation due to coke is reversible. Coking can be expected to deactivate the acid sites and then affect the hydrogen transfer reactions depending on acid site density to suppress undesired reactions. Meanwhile, it also has obvious influence on the pore structure of molecular sieve: (i) limitation of the access of n-heptane to the active sites, (ii) blockage of the access to the sites of the cavities (or channel intersections) in which the coke molecules are situated, (iii) blockage of the access to the sites of the pores in which there are no coke molecules [32-35]. However, pre-coking using carbonaceous compounds with large molecules is supposed that coke only deposit externally, the internal sites keep unchanged [36]. These coke molecules trapped in the zeolite micropores being relatively simple, are not generally inert with respect to the reactants or intermediates of the desired reactions and hence can significantly affect the activity and selectivity and be used as one of the surface modifications in some articles [37-40]. Bauer et al. suggested that

in terms of surface acidity inactivation. Samples modified by carbonaceous deposits were found to be more effective compared to those by one-cycle silica deposition: (i) pre-coked HZSM-5 showed a higher shape selectivity during xylene isomerization, (ii) pre-coking of HFER had no specific effect on isobutene selectivity [38]. While other researchers found that catalyst with 0.6 wt% coke deposition presented 20 times of the conversion on the fresh catalyst in chloromethane to olefins reaction, arising from the behaviour of coke working as an important reaction center for olefin assembly, which can eliminate the induction period of the reaction and govern the conversion and product selectivity [40].

Regarding of the pre-coking, it is believed that large amount of coke doesn't only selectively deposit on the outer surface of the catalyst, but also deactivate the internal active sites [41]. Thus, the pre-coking process should be carefully controlled and the grow rule of pre-coke species during the reaction should be mastered. Studies focusing on the stabilization or evolution of carbonaceous compounds on pre-coked catalysts during the reaction are few, thus the industrial application of pre-coking is worth deliberated.

In this study, various pre-coked catalysts were prepared with different reaction parameters: coking precursor, coking temperature and time. The textural properties of HZSM-5 catalyst before and after pre-coking have been characterized by series methods. The pre-coked catalysts were used in methanol to propylene reactions. The nature of coke species on the pre-coked ZSM-5 and the secondary-coked ZSM-5 in MTP conversions was examined by TGA method. Furthermore, the catalytic performances on propylene selectivity of these pre-coked catalysts have been investigated.

2. Experimental

2.1. Materials and pre-coking procedures

The fabrication process of shaped parent ZSM-5 catalyst involves two key steps: the preparation of zeolite powder and shaping process of binder/zeolite. The obtained product was cylindrical, with size of 1.2 mm-1.5 mm.

The ZSM-5 zeolite powder ($n_{\text{SiO}_2/\text{Al}_2\text{O}_3}=180$) was bought from Suzhou Zhi hydrocarbon New Material Technology Co., Ltd. This powder was washed and filtrated repeatedly for three times and then dried at 110 °C overnight. Finally, the product was calcined at 550 °C for 6 h in air.

Zeolite powder (350 g), binders (150 g, 75 wt% boehmite), sesbania powder (9 g), nitric acid (18 g, 66 wt%) and deionized water (180 g) were strongly blended at ambient temperature to form a homogeneous mixture, shaped into a uniform body by rapid extrusion molding and then dried at 120 °C for 12 h. The product was denoted as C-0.

Catalysts used methanol (99.9 wt%) and 1-hexene (99.9 wt%) as the pre-coke precursors were named M-x-y and H-x-y, respectively, while catalysts used methanol and deionized water as the pre-coking feeding were named nM-W-x-y, x is representing for the coking time (h), y is meaning for the coking temperature (°C), n is the mass percent of methanol in the methanol and deionized water solution (%). Water is very important in MTP reaction. And during the reaction, MeOH produces water. The water produced by MeOH can also reduce partial pressure of methanol, remove heat of reaction and inhibit the formation of carbon deposits on Catalysts.

The pre-coking ZSM-5 samples were prepared under different coking precursors (methanol, 1-hexene and methanol & deionized water) with a weight hour space velocity (WHSV) of 1.0 h⁻¹, different temperature (250 °C & 480 °C) and time (8 h & 35 h) in a stainless-steel tube fix-bed reactor (i.d. 12 mm). Before entering the reactor, the coking precursors needs to be metered by the pump, then heated and gasified in turn. Before and after the pre-coking procedures, the catalysts were dried under the atmosphere of N₂ at 120 °C for 3 h.

2.2. Characterization of catalysts

The BET surface area and porosity of the parent ZSM-5 and pre-coked catalysts were measured by N₂ adsorption and desorption method (Micromeritics ASAP 2400), the external surface area was tested by t-plot, and the mesopore volume was evaluated from the total pore volume subtracted the micropore volume. Prior to all measurement, the catalyst was outgassed at 120 °C for 5 h in order to desorb water.

Temperature Programmed Desorption with ammonia gas (NH₃-TPD) was used to examine the acid sites amount and distribution. **The catalyst was heated at 550 °C for 1 h under helium atmosphere to remove physical absorbed water, and then cooled to 100 °C and kept for 30 min for ammonia adsorption.** The physical adsorbed ammonia was removed by blowing helium for 1 h, and then the catalyst was carried out to desorb chemical absorbed ammonia by increasing the temperature linearly from 100 °C to 550 °C with heating rate of 10 °C min⁻¹. **In order to ensure the repeatability of data, all the operations are carried out under helium atmosphere.**

Pre-coked catalysts were characterized by elementary analysis (Thermo Finnigan Flash EA 1112) and Thermo Gravimetric Analyzer (TGA). The catalyst was placed in a crucible, lifted in the middle of the reactor tube. Before testing, the carrier gas N₂ was introduced at 120 °C with a flowing rate of 30 mL min⁻¹ for 4 h to remove the absorbed water in catalyst. Then the carrier gas was replaced by a mixture of O₂ (80 vol %) diluted in nitrogen with a total flowing rate of 50 mL min⁻¹, and the reactor temperature was increased from 120 °C to 850 °C with a heating rate of 10 °C min⁻¹.

2.3. Catalytic measurements

The catalytic testing was performed in a stainless-steel tube fix-bed reactor (i.d. 12 mm). 5 g of the ZSM-5 zeolite catalyst was mixed with 5 g glass beads. The reaction was carried out at 480 °C, and a WHSV of 1.0 h⁻¹. The methanol partial pressure was kept at 24 kPa by diluent H₂O. The product was analyzed by gas chromatography with PLOT-Q column (30 m x 0.32 mm x 25 μm) from 120 °C (keep 3 min) to 250 °C (keep 15 min) with a heating rate of 20 °C min⁻¹.

Table 1. Textural properties of parent and pre-coked ZSM-5.

sample	S _{surf} /(m ² ·g ⁻¹)			Pore volume/(cm ³ ·g ⁻¹)		
	BET ^a	micro ^b	external ^b	micro ^b	meco ^c	total ^d
C-0	349	302	47	0.146	0.231	0.377
1M-W-35-480	321	276	45	0.140	0.250	0.390
5M-W-35-480	319	271	48	0.135	0.235	0.370
M-8-480	329	279	50	0.137	0.237	0.374
M-35-480	318	276	42	0.137	0.225	0.362
M-35-250	340	293	47	0.142	0.232	0.374
H-8-250	538	445	93	0.219	0.408	0.627
H-8-480	333	291	42	0.140	0.229	0.369

S_{surf} : specific surface area.

BET^a : multi-point BET surface area.

^b from t-plot method.

^c from BJH method.

^d at P/P0≈0.98.

3. Results and Discussion

3.1. Physical characterization of the catalysts

The textural properties of pre-coked and parent catalysts are listed in Table 1. It can be seen that most of the catalysts surface area and micropore volume are decreased after pre-coking, except for sample H-8-250, whose pre-coking species came from the conversion of 1-hexene at 250 °C. For pre-coked samples (1M-W and 5M-W) prepared with abundant of water at high temperature (480 °C), an increasing trend of mesopore volume can be seen as the water concentration increased, suggesting the dealumination effect of steam on the catalysts. While the effect of coke on blocking the micropores can be observed as the reduction trend of micropore volume didn't follow as the stream content. **The pore volume of pre-coked sample 5M-W-35-480 is lower than sample 1M-W-35-480, and is similar with present sample C-0.**

For pre-coked catalysts using methanol as coke precursor, the external surface is decreased notably as the coking time extended from 8 h to 35 h at 480 °C, while no distinguish trend can be seen in the micropore volume, indicating that the pre-coke species mainly deposited on the outer surface of the particles. For pre-coked sample (M-35-250) prepared at low temperature (250 °C), a lesser destruction of micropores is observed, and the external surface and mesopore volume are nearly no change, which can be attribute to the low temperature limited the forming of carbonaceous compounds.

Catalyst using 1-hexene as pre-coke species has lower external surface compared with catalyst using methanol as coke precursor at the same reaction conditions. This is due to the fact that molecule with big size is more likely to condensed and deposited on the external of the zeolite [42].

3.2. Acid properties of the pre-coked catalysts

The NH₃-TPD profiles obtained from parent and pre-coked ZSM-5 samples are shown in Figure 1. Two broad desorption peaks maxima at 202 and 315 °C are assignable to weak acid site caused by ammonium cations and strong acid sites, respectively [43]. The desorption peaks in pre-coked sample H-8-250 are no difference with present ZSM-5, indicating that carbonaceous compounds formed by 1-hexene at low temperature (250 °C) are really unstable, which can be easily decomposed at high temperature, thus this kind of pre-coke species had no effect on the acidity of catalysts.

The NH₃-TPD results reveal only the difference in the amount of acid sites but not in their strength of the pre-coked catalysts. The acidity distribution of sample M-35-250 and H-8-480 are almost the same, though they were prepared under quite different pre-coking conditions. Similar finding was obtained from the samples 5M-W-35-480 and M-35-480. This observation indicates that different pre-coking parameters coordinated are possible to result in the same modification of the acidity.

3.3. Coke characterization of the pre-coked catalysts before and after MTP reaction

The amount of oxidable coke was determined by TGA, and the results of these pre-coked catalysts are shown in Figure 2 and Figure 3. It can be seen in Figure 2, that burning off the pre-coke formed at high temperature (480 °C) reveals a combustion peak at 600 °C, which is attributed to the combustion of polyaromatics species. While the pre-coked sample M-35-480 is the exception, as two combustion peaks can be observed in Figure 2d, revealing two types of coke species [44], coke I (600 °C) and coke II (670 °C), and the amount of these coke species are exchanged after MTP reaction. This observation can be ascribing to the different coke locations of the pre-coke and secondary-coke [45]. Pre-coke on sample

M-35-480 is deposited mainly on the external surface, while on the effect of stream during the MTP reaction, part of the pre-coke decomposed and reassembled at the internal pores [17], thus increasing the proportion of internal coke.

Figure 1. NH₃-TPD profiles of parent and pre-coked samples: (a) 1M-W-35-480; (b) 5M-W-35-480; (c) M-8-480; (d) M-35-480; (e) M-35-250; (f) H-8-250; (g) H-8-480; (h) C-0 (present ZSM-5).

The combustion peak of pre-coke formed at low temperature (250 °C) is centered on 300 °C (Figure 2e and Figure 2f), containing mainly the olefinic compounds [46]. This kind of coke species are condensed and rearranged via hydrogen transfer steps at high temperature (480 °C), and resulting in the formation of polyaromatics. This fact reflects that coke formed at low temperature is of olefinic nature, but the stable form of coke at high temperature is polyaromatics, which is highly condensed and dehydrogenized of olefinic or aliphatic species.

Table 2. Elementary analysis of carbonaceous compounds of pre-coked catalysts before and after MTP reaction.

Samples	C/H mass ratio of pre-coke	C/H mass ratio of secondary-coke
M-35-480	21.30	27.79
H-8-250	8.61	14.50
C-0	0	5.19

Figure 2. DTG profiles of pre-coked catalysts before (solid line) and after (dash line) MTP reaction: (a) 1M-W-35-480; (b) 5M-W-35-480; (c) M-8-480; (d) M-35-480; (e) M-35-250; (f) H-8-250; (g) H-8-480; (h) C-0 (present ZSM-5).

The elementary analysis of carbonaceous compounds of these abnormal pre-coked samples is shown in Table 2. It can be seen that pre-coke formed at low temperature (250 °C) via 1-hexene is of more olefinic nature, while the coke species turned to be of more aromatic nature after MTP reaction, this fact verified that coke formed at low temperature is unstable at high temperature [47], which will be transformed to polyaromatics. While pre-coke formed at severely high temperature via methanol conversion is more aromatic and condensed. This type of pre-coke is quite stable at high temperature, but continues to dehydrogenize and condense into carbonaceous compounds with more big size.

Figure 3 shows the coke content of pre-coked catalysts before and after MTP reaction. It can be seen that the coke content of pre-coked sample 5M-W-35-480 is obviously lower than present catalysts after MTP conversions. Secondary coke content among samples 1M-W-35-480, M-8-480, M-35-250 and present sample C-0 are no distinguishable difference, while the others (M-35-480, H-8-250, H-8-480) are notably higher than sample C-0. This fact suggests that different pre-coking conditions lead to the different distribution of coke deposits, furthermore results in the different textural properties of the catalysts, taking charge of the various coke performances.

Pre-coking via 1-hexene conversion obtains higher coke deposition compared with using methanol as pre-coke precursor, this can be attributed to the dehydration of methanol, which generating the equal moles of water, and then suppressed the formation of coke [48]. While no water is appeared at the conversion of 1-hexene, thus the pre-coke in the pores or on the outer surface acts as coke-nucleus and accelerates the formation of secondary-coke.

Combing the pre-coking treatment with streaming is very effective to inhibit the secondary-coke formation, as both coke and water can used to modify the pore structure and acidity of the catalyst [49], while the proportions of these two factors need to be optimized. It can be seen that pre-coking by 5% methanol and stream is more effective than 1% methanol and stream in suppressing the coke deposition. This can be clarified by the pre-coke distribution which was verified by the textural properties of these two pre-coked samples characterized by BET methods. Pre-coke is likely to fill in the mesopores formed by dealumination by stream, primarily, thus the acid sites near the mesopores wall are covered, and the diffusion resistance of molecules are strengthened by the steric hindrance of pre-coke, increasing the difficulty of macromolecular carbon deposition on the mesopores. Thus, suitable pre-coking process is instead done well in restraining coke deposition than hydrothermal treatment.

Figure 3. Comparison of pre-coke content and secondary-coke content of each sample.

3.4. Catalytic Properties of the pre-coked ZSM-5 in MTP reaction

The pre-coked catalysts were classified into two types according to the acidity of the catalyst, samples 1M-W-35-480, 5M-W-35-480 and M-35-480 were prepared at relatively severe coking conditions, the destruction of acid sites were more obviously, hence, these pre-coked catalysts were marked as type A, and the others were marked as type B. The conversion of methanol and selectivity of propylene are displayed against time on stream and the results are shown in Figure 4. The conversions of pre-coked catalysts are above 99%, except sample 1M-W-35-480, whose conversion decreased rapidly after 15 h of TOS. **The reason is there are too much water added during the preprocessing of the sample 1M-W-35-480. It causes serious dealumination when sample is under hydrothermal condition and cause many losses of acidic sites.**

The propylene selectivity of pre-coked catalysts (type A, Figure 4c) are substantially higher than pre-coked catalyst of type B (Figure 4d), the propylene selectivity of sample 5M-W-35-480 is reached to 42% after 15 h of TOS. This is due to the coordinated modification of coke and stream, resulting in a suitable pore structure for the production of propylene. As sample M-35-480 was severely coked, the diffusion limitation of reacting molecules was enhanced, not only restrained the production of macromolecules, but also affected the diffusion of micromolecules like propylene [50]. Pre-coking prepared at low temperature (250 °C) present no improvement in propylene selectivity, arising from the unstable form of the carbonaceous compounds.

Figure 4. Catalytic performances of pre-coked ZSM-5 in MTP reaction: (a and b) methanol conversion; (c and d) selectivity of propylene

4. Conclusion

The pre-coking modification of ZSM-5 zeolite has been performed to improve the propylene selectivity of the catalysts maintaining high activity. Detailed features of pre-coked catalysts have been obtained by the results of the characterization and catalytic investigation. Pre-coking had less destruction to the acid sites of ZSM-5 compared with steam, and retained the high activity of the catalyst. Pre-coke formed at low temperature (250 °C) was unstable, the olefinic and aliphatic compounds were transformed into polyaromatic species at high reacting temperature. While 5% pre-coking combining with 95% streaming treatment at high temperature (480 °C) was very effective to inhibit the coke formation during the MTP conversions, as the pre-coke primarily deposited on the mesopores formed by dealumination of the crystal, and the selectivity of propylene was reached to 42%.

Data Availability. This article does not contain any additional data.

Competing Interests. We declare we have no competing interests.

Authors' Contributions. L.W. designed this work, conducted the experimental operations and catalytic optimization, and drafted the manuscript; J.Q., H.J., L.A. and C.G. participated in the analysis and interpretation of data; X.Y., Z.J. and A.Z. revised the manuscript for important intellectual contents; D.L. ensured that the work was appropriately investigated and analysed. All authors gave final approval for publication.

Funding. This work was supported by the National International Cooperation S & T Project of China [grant number 2015DFA40660], Key R&D Programs of NingXia Province [grant number 2018BDE63020, 2018BDE63003, and 2018BDE63007]

Acknowledgements. We thank Prof. Dr. Siegfried Stapf for the analysis of data and revised the manuscript.

Notes and references

1. Zhao C, WACHS, Israel E. 2006 Selective oxidation of propylene to acrolein over supported V₂O₅/Nb₂O₅ catalysts: An in situ Raman, IR, TPSR and kinetic study. *Catal. Today*. **118**, 332-343. (doi:10.1016/j.cattod.2006.07.018)
2. Castonguay L A, Rappe A K. 1992 Ziegler-Natta catalysis. A theoretical study of the isotactic polymerization of propylene. *J. Am. Chem. Soc.* **114**, 5832-5842. (doi:10.1021/ja00040a053)
3. Xie J, Zhang Q, Chuan g KT. 2000 An IGC Study of Pd/SDB Catalysts for Partial Oxidation of Propylene to Acrylic Acid. *J. Catal.* **191**, 86-92. (doi:10.1006/jcat.1999.2796)
4. Muntasar A, Mao RLV, Yan HT. 2010 "Petroleum Gas Oil-Ethanol" Blends Used as Feeds: Increased Production of Ethylene and Propylene over Catalytic Steam-Cracking (CSC) Hybrid Catalysts. Different Behavior of Methanol in Blends with Petroleum Gas Oil. *Ind. Eng. Chem. Res.* **49**, 3611-3616. (doi:10.1021/ie902035m)
5. Abul-Hamayel MA, Aitani AM, Saeed MR. 2010 The effect of pre- coking on the activity and selectivity of the catalytic cracking of squalane. *J. Chem. Technol. Biotechnol.* **28**, 923-929. (doi:10.1002/jctb.1167)
6. Koempel H, Liebner W. 2007 Lurgi's methanol to propylene (MTP): report on a successful commercialisation. *Stud. Surf. Sci. Catal.* **167**, 261-267. (doi:10.1016/S0167-2991(07)80142-X)
7. Jiang B, Feng X, Yan L, Jiang Y, Liao Z, Wang J, Yang Y. 2014 Methanol to Propylene Process in a Moving Bed Reactor with Byproducts Recycling: Kinetic Study and Reactor Simulation. *Ind. Eng. Chem. Res.* **53**, 4623-4632. (doi:10.1021/ie500250d)
8. Zhang H, Ning Z, Liu H, Shang J, Han S, Jiang D, Yu J, Guo Y. 2017 Bi₂O₃ modification of HZSM-5 for methanol-to-propylene conversion: evidence of olefin-based cycle. *RSC Adv.* **7**, 16602-16607. (doi:10.1039/C6RA27849C)
9. Keil FJ. 1999 Methanol-to-hydrocarbons: process technology. *Micropor. Mesopor. Mater.* **29**, 49-66. (doi:10.1016/S1387-1811(98)00320-5)
10. Wang Y, Rui G, Zhang F, Gan Y, Zhang X. 2018 Catalyst with high C4 olefin selectivity for preparing olefin from methanol and preparation method thereof. US Patent. US09856183B2.
11. Haw JF, Song W, Marcus DM, Nicholas JB. 2003 The mechanism of methanol to hydrocarbon catalysis. *Accounts Chem. Res.* **36**, 317-326. (doi:10.1021/ar020006o)
12. Bjørgen M, Svelle S, Joensen F, Nerlov J, Kolboe S, Bonino F, Palumbo L, Bordiga S, Olsbye U. 2007 Conversion of methanol to hydrocarbons over zeolite H-ZSM-5: On the origin of the olefinic species. *J Catal.* **249**, 195-207. (doi:10.1016/j.jcat.2007.04.006)
13. Stöcker M. 1999 Methanol-to-hydrocarbons: catalytic materials and their behavior. *Micropor. Mesopor. Mater.* **29**, 3-48. (doi:10.1016/S1387-1811(98)00319-9)
14. Mikkelsen Ø, Kolboe S. 1999 The conversion of methanol to hydrocarbons over zeolite H-beta. *Micropor. Mesopor. Mater.* **29**, 173-184. (doi:10.1016/S1387-1811(98)00329-1)
15. Derouane EG, Nagy JB, Dejaifve P, Hooff JHCV, Spekman BP, Védrine JC, Naccache C. 1978 Elucidation of the mechanism of conversion of methanol and ethanol to hydrocarbons on a new type of synthetic zeolite. *J Catal.* **53**, 40-55. (doi:10.1016/0021-9517(78)90006-4)
16. Schulz H. 2010 "Coking" of zeolites during methanol conversion: Basic reactions of the MTO-, MTP- and MTG processes. *Catal. Today*. **154**, 183-194. (doi:10.1016/j.cattod.2010.05.012)
17. Mores D, Kornatowski J, Olsbye U, Weckhuysen BM. 2015 Coke formation during the methanol-to-olefin conversion: in situ microspectroscopy on individual H-ZSM-5 crystals with different Brønsted acidity. *Chem.* **17**, 2874-2884. (doi:10.1002/chem.201002624)
18. Sun C, Yang ., Du J, Qin F, Liu Z, Shen W, Xu H, Tang Y. 2012 Dehydrogenation inhibition on nano-Au/ZSM-5 catalyst: a novel route for anti-coking in methanol to propylene reaction. *Chem. Commun.* **48**, 5787-5789. (doi:10.1039/c2cc30607g)
19. Chen JQ, Bozzano A, Glover B, Fuglerud T, Kvisle S. 2005 Recent advancements in ethylene and propylene production using the

- UOP/Hydro MTO process. *Catal. Today*. **106**, 103-107. (doi:10.1016/j.cattod.2005.07.178)
20. Mei C, Wen P, Liu Z, Liu H, Wang Y, Yang W, Xie Z, Hua W, Gao Z. 2008 Selective production of propylene from methanol: Mesoporosity development in high silica HZSM-5. *J Catal.* **258**, 243-249. (doi:10.1016/j.jcat.2008.06.019)
21. Mante OD, Agblevor FA, Oyama ST, Mcclung R. 2012 The effect of hydrothermal treatment of FCC catalysts and ZSM-5 additives in catalytic conversion of biomass. *Appl. Catal. A: Gen.* **445-446**, 312-320. (doi:10.1016/j.apcata.2012.08.039)
22. Beyer HK. 2002 Dealumination Techniques for Zeolites. In: Post-Synthesis Modification I. *Springer*. **3**, 203-255. (doi:10.1007/3-540-69750-0_3)
23. Topsøe NY, Joensen F, Derouane EG. 1988 IR studies of the nature of the acid sites of ZSM-5 zeolites modified by steaming. *J Catal.* **110**, 404-406. (doi:10.1016/0021-9517(88)90330-2)
24. Rhee KH, Rao VUS, Stencil JM, Melson GA, Crawford JE. 1983 Supported transition metal compounds: Infrared studies on the acidity of Co/ZSM-5 and Fe/ZSM-5 catalysts. *Zeolites*. **3**, 337-343. (doi:10.1016/0144-2449(83)90179-3)
25. Benito PL, Gayubo AG, Aguayo AT, Olazar M, Bilbao J. 2015 Effect of Si/Al Ratio and of Acidity of H-ZSM5 Zeolites on the Primary Products of Methanol to Gasoline Conversion. *J. Chem. Technol. Biotechnol.* **66**, 183-191. (doi:10.1002/(SICI)1097-4660(199606)66:2<183::AID-JCTB487>3.0.CO;2-K)
26. Corma A, Fornés V, Rey F. 1990 Extraction of extra-framework aluminium in ultrastable Y zeolites by (NH₄)₂SiF₆ treatments: I. Physicochemical Characterization. *Appl. Catal.* **59**, 267-274. (doi:10.1016/S0166-9834(00)82203-4)
27. Fleisch TH, Meyers BL, Ray GJ, Hall JB, Marshall CL. 1986 Hydrothermal dealumination of faujasites. *J Catal.* **99**, 117-125. (doi:10.1016/0021-9517(86)90205-8)
28. Palborbely G, Beyer HK. 2003 Isomorphous solid-state substitution of Si for framework Al in Y zeolite using crystalline (NH₄)₂[SiF₆] as reactant. *Phys. Chem. Chem. Phys.* **5**, 2145-2153. (doi:10.1039/b211763k)
29. Gayubo AG, Aguayo AT, Olazar M, Vivanco R, Bilbao J. 2003 Kinetics of the irreversible deactivation of the HZSM-5 catalyst in the MTO process. *Chem. Eng. Sci.* **58**, 5239-5249. (doi:10.1016/j.ces.2003.08.020)
30. Corma A, Mengual J, Miguel PJ. 2012 Steam catalytic cracking of naphtha over ZSM-5 zeolite for production of propene and ethene: Micro and macroscopic implications of the presence of steam. *Appl. Catal. A: Gen.* **417-418**, 220-235. (doi:10.1016/j.apcata.2011.12.044)
31. Lucas AD, Canizares P, Durán A, Carrero A. 1997 Dealumination of HZSM-5 zeolites: Effect of steaming on acidity and aromatization activity. *Appl. Catal. A: Gen.* **154**, 221-240. (doi:10.1016/S0926-860X(96)00367-5)
32. Abbot J, Wojciechowski BW. 1988 ChemInform Abstract: The Effect of Temperature on the Product Distribution and Kinetics of Reactions of n-Hexadecane on HY Zeolite. *Cheminform.* **109**, 274-283. (doi:10.1002/chin.198823100)
33. Chia D, Trimm D. 2010 The effect of pre-coking on the activity and selectivity of the catalytic cracking of squalane. *J. Chem. Technol. Biotechnol.* **80**, 353-355. (doi:10.1002/jctb.1167)
34. Guisnet M. 2009 Prevention of zeolite deactivation by coking. *J. Mol. Catal. A: Chem.* **305**, 69-83. (doi:10.1016/j.molcata.2008.11.012)
35. Guisnet M, Magnoux P. 1989 Coking and deactivation of zeolites. Influence of the Pore Structure. *Appl. Catal.* **54**, 1-27. (doi:10.1016/S0166-9834(00)82350-7)
36. Tsai TC, Liu SB, Wang I. 1999 Disproportionation and transalkylation of alkylbenzenes over zeolite catalysts. *Appl. Catal. A: Gen.* **181**, 355-398. (doi:10.1016/S0926-860X(98)00396-2)
37. Takamitsu Y, Yamamoto K, Yoshida S, Ogawa H, Sano T. 2014 Effect of crystal size and surface modification of ZSM-5 zeolites on conversion of ethanol to propylene. *J Porous. Mat.* **21**, 433-440. (doi:10.1007/s10934-014-9789-4)
38. Bauer F, Chen WH, Bilz E, Freyer A, Sauerland V, Liu SB. 2007 Surface modification of nano-sized HZSM-5 and HFER by pre-coking and silanization. *J Catal.* **251**, 258-270. (doi:10.1016/j.jcat.2007.08.009)
39. Guisnet M. 2002 "Coke" molecules trapped in the micropores of zeolites as active species in hydrocarbon transformations. *J. Mol. Catal. A: Chem.* **182**, 367-382. (doi:10.1016/S1381-1169(01)00511-8)
40. Wei Y, Zhang D, Chang F, Xia Q, Su BL, Liu Z. 2009 Ultra-short contact time conversion of chloromethane to olefins over pre-coked SAPO-34: direct insight into the primary conversion with coke deposition. *Chem. Commun.* **34**, 5999-6001. (doi:10.1039/B909218H)
41. Bauer F, Bilz E, Freyer A. 2005 C-14 studies in xylene isomerization on modified HZSM-5. *Appl. Catal. A: Gen.* **289**, 2-9. (doi:10.1016/j.apcata.2005.04.008)

42. Uguina MA, Serrano DP, Grieken RV, Vènes S. 1993 Adsorption, acid and catalytic changes induced in ZSM-5 by coking with different hydrocarbons. *Appl. Catal. A: Gen.* **99**, 97-113. (doi:10.1016/0926-860X(93)80093-6)
43. Kubo K, Iida H, Namba S, Igarashi A. 2014 Effect of steaming on acidity and catalytic performance of H-ZSM-5 and P/H-ZSM-5 as naphtha to olefin catalysts. *Micropor. Mesopor. Mater.* **188**, 23-29. (doi:10.1016/j.micromeso.2014.01.002)
44. Guisnet M, Magnoux P, Martin D. 1997 Roles of acidity and pore structure in the deactivation of zeolites by carbonaceous deposits. *Stud. Surf. Sci. Catal.* **111**, 1-19. (doi:10.1016/S0167-2991(97)80138-3)
45. Epelde E, Ibañez M, Aguayo AT, Gayubo AG, Bilbao J, Castaño P. 2014 Differences among the deactivation pathway of HZSM-5 zeolite and SAPO-34 in the transformation of ethylene or 1-butene to propylene. *Micropor. Mesopor. Mater.* **195**, 284-293. (doi:10.1016/j.micromeso.2014.04.040)
46. Wang J, Hassan F, Chigada PI, Rigby SP, Alduri B, Wood J. 2009 Deactivation during 1-Hexene Isomerization over Zeolite Y and ZSM5 Catalysts under Supercritical Conditions. *Ind. Eng. Chem. Res.* **48**, 7899-7909. (doi:10.1021/ie101876f)
47. Pradhan AR, Lin T., Chen W, Jong S, Wu J, Chao K, Liu S. 1999 EPR and NMR Studies of Coke Induced Selectivation over H-ZSM-5 Zeolite during Ethylbenzene Disproportionation Reaction. *J Catal.* **184**, 29-38. (doi:10.1006/jcat.1999.2431)
48. Gayubo AG, Aguayo AT, Atutxa A, Prieto R, Bilbao J. 2004 Role of Reaction-Medium Water on the Acidity Deterioration of a HZSM-5 Zeolite. *Ind. Eng. Chem. Res.* **43**, 5042-5048. (doi:10.1021/ie0306630)
49. Kim YH, Lee KH, Lee JS. 2011 The effect of pre-coking and regeneration on the activity and stability of Zn/ZSM-5 in aromatization of 2-methyl-2-butene. *Catal. Today.* **178**, 72-78. (doi:10.1016/j.cattod.2011.07.002)
50. Song C, Liu K, Zhang D, Liu S, Li X, Xie S, Xu L. 2014 Effect of cofeeding n-butane with methanol on aromatization performance and coke formation over a Zn loaded ZSM-5/ZSM-11 zeolite. *Appl. Catal. A: Gen.* **470**, 15-23. (doi:10.1016/j.apcata.2013.10.036)